# Life’s Simple 7 and Risk of Peripheral Artery Disease: Results from the PREDIMED Study and an Updated Meta-Analysis

**DOI:** 10.3390/nu17132058

**Published:** 2025-06-20

**Authors:** Nieves López-Laguna, Estefanía Toledo, María S. Hershey, Nancy Babio, José V. Sorlí, Emilio Ros, Miguel Ángel Muñoz, Ramón Estruch, José Lapetra, Carlos Muñoz-Bravo, Miquel Fiol, Inmaculada Bautista-Castaño, Xavier Pinto, Carolina Ortega-Azorín, Javier Hernando-Redondo, Jordi Salas-Salvadó, Lucas Tojal-Sierra, Miguel A. Martínez-González, Miguel Ruiz-Canela

**Affiliations:** 1Department of Preventive Medicine and Public Health, School of Medicine, Navarra Institute for Health Research (IdiSNA), University of Navarra, 31008 Pamplona, Spain; nlopezl@unav.es (N.L.-L.); etoledo@unav.es (E.T.); mhershey@fas.harvard.edu (M.S.H.); 2Emergency Department, Clínica Universidad de Navarra, 28027 Madrid, Spain; 3CIBER Fisiopatología de la Obesidad y Nutrición (CIBEROBN), Instituto de Salud Carlos III (ISCIII), 28029 Madrid, Spain; nancy.babio@urv.cat (N.B.); jose.sorli@uv.es (J.V.S.); erosr@recerca.clinic.cat (E.R.); restruch@ub.edu (R.E.); joselapetra543@gmail.com (J.L.); miguel.fiol@ssib.es (M.F.); inmaculada.bautista@ulpgc.es (I.B.-C.); xpinto@bellvitgehospital.cat (X.P.); carolina.ortega@uv.es (C.O.-A.); jhernando1@researchmar.net (J.H.-R.); jordi.salas@urv.cat (J.S.-S.); lutojal@hotmail.com (L.T.-S.); 4Human Flourishing Program, Institute for Quantitative Social Science, Harvard University, Boston, MA 02138, USA; 5Universitat Rovira i Virgili, Departament de Bioquímica i Biotecnologia, Unitat de Nutrició Humana, Grup Almentació, Nutrició, Desenvolupament i Salut Mental (ANUT-DSM), 43880 Reus, Spain; 6Institut D’Investigació Sanitària Pere Virgili (IISPV), 43880 Reus, Spain; 7Department of Preventive Medicine, University of Valencia, 46010 Valencia, Spain; 8Institut d’Investigacions Biomédiques August Pi Sunyer (IDIBAPS), Hospital Clínic, 08002 Barcelona, Spain; 9Instituto de Investigación en Atención Primaria Idiap Jordi Gol, 08002 Barcelona, Spain; mamunoz.bcn.ics@gencat.cat; 10Department of Internal Medicine, Institut d’Investigacions Biomédiques August Pi Sunyer (IDIBAPS), Hospital Clinic, University of Barcelona, 08002 Barcelona, Spain; 11Department of Family Medicine, Research Unit, Distrito Sanitario Atención Primaria Sevilla, 41001 Sevilla, Spain; 12Department of Public Health and Psychiatry, University of Malaga, 29015 Malaga, Spain; carlosmb@uma.es; 13Instituto de Investigación Biomédica de Málaga y Plataforma en Nanomedicina (IBIMA Plataforma BIONAND), 29015 Malaga, Spain; 14Health Research Institute of the Balearic Islands (IdISBa), 07240 Illes Balears, Spain; 15Research Institute of Biomedical and Health Sciences, University of Las Palmas de Gran Canaria, 35001 Las Palmas, Spain; 16Internal Medicine Department, Hospital Universitari de Bellvitge-IDIBELL, Universidad de Barcelona, 08097 Barcelona, Spain; 17Registre Gironí del Cor (REGICOR) Study Group, Hospital del Mar Research Institute (IMIM), 08097 Barcelona, Spain; 18Bioaraba Health Research Institute, Osakidetza Basque Health Service, Araba University Hospital, University of the Basque Country UPV/EHU, 01009 Vitoria-Gasteiz, Spain; 19Department of Nutrition, Harvard TH Chan School of Public Health, Boston, MA 02115, USA

**Keywords:** peripheral artery disease, Life’s Simple 7, cardiovascular health, Mediterranean diet, PREDIMED study, prevention

## Abstract

Background: Peripheral artery disease (PAD) is a major vascular condition often overlooked in prevention strategies. We aimed to evaluate the association between cardiovascular health, measured by Life’s Simple 7 (LS7), and the risk of PAD in a high-risk Mediterranean population. Methods: This prospective analysis included 7122 participants from the PREDIMED study (Prevención con Dieta Mediterránea) at high cardiovascular risk but free of cardiovascular disease at baseline. LS7 scores (0–14 points) were calculated using seven metrics: smoking status, body mass index, physical activity, blood pressure, total cholesterol, glucose metabolism, and adherence to the Mediterranean diet. Participants were categorized into inadequate (0–5), average (6–8), and optimal (9–14) cardiovascular health. Multivariable Cox regression models and Nelson–Aalen curves assessed the association between LS7 and PAD incidence over a median 4.8-year follow-up. A meta-analysis combining these results with three prior studies was also performed. Results: A total of 87 incident PAD cases were identified. Compared to participants with inadequate cardiovascular health, those with average and optimal LS7 scores exhibited significantly lower PAD risk (adjusted hazard ratio [HR] 0.37; 95% confidence interval [CI]: 0.22–0.61, and HR 0.25; 95% CI: 0.10–0.65, respectively). Each one-point increase in the LS7 score (range 0 to 14) was associated with an 22% lower PAD risk (HR 0.78; 95% CI: 0.68–0.90). The meta-analysis yielded a pooled HR of 0.81 (95% CI: 0.76–0.87), confirming consistent inverse associations across populations. Conclusions: Greater adherence to LS7 metrics is associated with a significantly reduced risk of PAD in high-risk Mediterranean individuals. Promoting LS7 adherence may represent an effective strategy for preventing both cardiovascular disease and PAD.

## 1. Introduction

Peripheral artery disease (PAD) is a term used to describe ischemia of the blood vessels supplying the lower limbs. The prevalence of PAD in people aged 40 years and older was 113 million in 2019 [1], and around 74,000 deaths were caused by this disease [2]. The major known risk factors are diabetes, hypertension, hypercholesterolemia, and smoking [3,4,5]. It is also known that the combined effect of these factors greatly increases the risk of PAD [6]. Therefore, a primary prevention strategy for PAD should be focused on promoting overall healthy lifestyles [7].

In 2010, the American Heart Association (AHA) designed the Life’s Simple 7 (LS7) metric to improve the cardiovascular health of all Americans by 20% in 2020 [8] Ideal cardiovascular health was defined by the presence of seven metrics; they include four health behaviors (low BMI, avoiding tobacco, healthy diet, and physical activity) and three health factors (cholesterol, blood pressure, and blood glucose). These seven metrics are total cholesterol <200 mg/dL, systolic blood pressure <120 mmHg, diastolic blood pressure <80 mmHg, baseline blood glucose <100 mg/dL, physical activity at goal levels, ideal diet (consistent with current guideline recommendations), nonsmoking, and BMI <25 kg/m^2^ [8].

LS7-defined ideal CV health adherence has been associated with lower risks of cardiovascular disease. A recent systematic review and meta-analysis with three studies observed a reduction of 80% (95% confidence interval, 46–92%) of PAD risk in people following an ideal LS7 metric compared to an inadequate LS7 metric [9]. However, no study has been published in a Mediterranean region or incorporating the Mediterranean diet as an ideal model of a healthy diet.

In the PREDIMED trial, we observed a 66% (47–79%) reduction of other cardiovascular diseases different from PAD (myocardial infarction, stroke, or cardiovascular death) in participants with ≥4 LS7 metrics compared with participants with only 0 to 1 metrics [10]. Our goal was to analyze the relationship between a greater number of metrics in LS7 and the development of PAD in a population with high cardiovascular risk from the PREDIMED study, and to update the previous meta-analysis.

## 2. Materials and Methods

### 2.1. Study Population

The PREDIMED study (Prevención con Dieta Mediterranean) is a parallel, multicenter, randomized trial conducted in Spain, designed for studying the relation between the Mediterranean diet and cardiovascular disease (http://www.isrctn.com/ISRCTN35739639, accessed on 15 June 2025). The design, description of the methods, and objectives can be found in depth elsewhere [11,12]. The trial recruited participants in 11 centers. They were men (between 55 and 80 years of age) and women (between 60 and 80 years of age) without cardiovascular disease at baseline who had type 2 diabetes mellitus or at least three of the following risk factors: smoking, hypertension, elevated low-density lipoprotein cholesterol levels, low high-density lipoprotein cholesterol levels, overweight or obesity, or a family history of premature coronary heart disease. Participants were randomly assigned 1:1:1 to one of three diets (a Mediterranean diet supplemented with extra virgin olive oil, a Mediterranean diet supplemented with nuts, or a low-fat diet in the control group).

Of the 7447 participants recruited in the PREDIMED study between 2003 and 2009, we excluded 12 participants with prevalent PAD at baseline, 208 without available follow-up, and 105 participants with a total daily energy intake outside predefined limits (less than 800 or more than 4200 Kcal/day among men and less than 600 or more than 3600 Kcal/day among women), leaving a total of 7122 participants for this analysis.

### 2.2. End-Point

The primary end-point in this study was the development of symptomatic PAD. Medical records were examined, blinded to the exposure, by the Clinical Event Adjudication Committee to document the presence of PAD diagnosis if participants reported a disorder of the lower extremity arteries during follow-up. The information to adjudicate new cases of PAD included ankle-brachial index measurements, treadmill exercise test, limb segmental pressure measurements, pulse volume recordings and imaging tests (duplex ultrasonography, magnetic resonance angiography, computed tomographic angiography or catheter-based radiocontrast angiography). Confirmed PAD cases required at least one of the following criteria: an ABI lower than 0.9 at rest, objective evidence of arterial occlusive disease, or an endovascular or open surgical revascularization (or amputation). All diagnoses of PAD were confirmed by an end-point adjudication committee who were blinded to the intervention group [13].

### 2.3. Life’s Simple 7

Cardiovascular health was assessed using the American Heart Association’s Life’s Simple 7 (LS7) metrics [8]. The total LS7 score (0–14 points) was calculated by summing the points from all seven components (Table 1), with each component contributing 0–2 points. Higher scores indicate better cardiovascular health. The healthy diet metric was defined by reaching at least 9 points on a validated scale of 14 points of adherence to the Mediterranean diet (MEDAS) [14]. Based on the total score, participants were classified into three cardiovascular health categories: (1) Poor cardiovascular health: 0–5 points, (2) Intermediate cardiovascular health: 6–8 points, and (3) Ideal cardiovascular health: 9–14 points.

### 2.4. Other Covariates

Analyses and measurements were performed during the baseline interview to assess hypertension, diabetes, and hypercholesterolemia. In addition, a physical examination was performed during which constants, such as blood pressure, measured in triplicate with an electronic device, and other variables such as weight, waist circumference, height, medication used, diet using a validated questionnaire [15], and physical activity were collected by trained personnel.

### 2.5. Statistical Analysis

Baseline characteristics are presented using the mean and standard deviation for the quantitative variables and proportions for the qualitative variables.

Each metric was assigned 0, 1, or 2 points, depending on the degree of adaptation to Life’s Simple 7 (Table 1). Adding these variables, each participant could have a result from 0 to 14 points. For the main analysis, participants were divided into 4 categories (0–1, 2, 3, and 4 or more metrics) based on the number of metrics they accumulated (inadequate, 0–5 points; average, 6–8 points; or optimal, 9–14 points).

To estimate the cumulative incidence of PAD according to LS7 categories, we calculated Nelson–Aalen cumulative hazard functions. To minimize potential selection bias and confounding effects, inverse probability weighting (IPW) was applied to the analysis. Specifically, stabilized weights were derived from logistic regression models that included age and sex as covariates.

The follow-up time of each participant was calculated as the time elapsed in years from the date of the initial questionnaire to the date of diagnosis of PAD, date of death, date of the last visit, or date of end of the study.

The analysis used an initial multivariate Cox regression model to calculate the hazard ratio (HR) and its 95% confidence interval (CI) adjusting for age, sex, educational level (primary or lower education, secondary, or university education), and alcohol intake (grams/day). In a second multivariable model, we additionally adjusted for height (meters), waist circumference (cm), family history of coronary heart disease, diabetes status, use of statins, and total energy intake. As an ancillary analysis, we also adjusted for type of work (employed, unpaid domestic work, retired, and other). The group with inadequate LS7 was used as a reference category.

A bibliographic search was conducted in PubMed using the terms “peripheral artery disease”, “Life’s Simple 7”, and “ideal cardiovascular health”. We identified the same articles included in a previous systematic review and meta-analysis [9], and all subsequent citations of these articles were reviewed [16,17,18,19]. A random-effects meta-analysis was conducted using the DerSimonian–Laird method to assess the pooled HR for the association between LS7 (as a continuous variable) and PAD. The HR per-point increase in LS7 was used to improve comparability across studies. Heterogeneity was evaluated using Cochran’s Q test and quantified using the I^2^ statistic. We also conducted a leave-one-out sensitivity analysis to evaluate the contribution of each study to the overall heterogeneity.

All statistical analyses were performed using Stata version 17.0 (StataCorp, College Station, TX, USA).

### 2.6. Ethics

In accordance with the Declaration of Helsinki (1975, revised in 2013), the PREDIMED trial was approved by the Research Ethics Committees of all recruiting centers (approval code: 50/2005; approval date: 5 July 2005). All participants received adequate written information and provided written informed consent before randomization. This trial was registered at controlled-trials.com (ISRCTN35739639).

## 3. Results

A total of 7122 participants were included in this study. The mean age was 67.1 years (SD 6.2) and 57.5% were women. The mean Life’s Simple 7 (LS7) score was 6.6 (SD 1.8), with values ranging from 0 to 12 points. The general characteristics of the population, according to the number of LS7 metrics, are shown in Table 2.

As expected, participants with average or optimal cardiovascular health (higher LS7 scores) showed better cardiovascular risk profiles, including lower blood pressure, glucose levels, BMI, and waist circumference, along with significantly higher physical activity levels, better dietary quality (higher MEDAS), and lower smoking prevalence, compared to those with inadequate scores. Additionally, average or optimal cardiovascular health was associated with a higher percentage of women, lower alcohol intake, and slightly higher energy intake.

A total of 87 incident PAD cases were identified after a median follow-up of 4.8 years. Figure 1 shows the Nelson–Aalen cumulative hazard curves for PAD (sex- and age-adjusted) according to LS7 categories over 5 years of follow-up. Participants with inadequate cardiovascular health (scoring 0 to 5 in LS7) demonstrated a progressively higher cumulative hazard of PAD as compared to those with average/optimal cardiovascular health (scoring >5 in LS7). By the end of the follow-up period, the cumulative incidence of PAD was 0.022 (2.2%) in the inadequate LS7 group versus 0.009 (0.9%) in the average/optimal LS7 group.

Table 3 shows that participants with average and optimal LS7 scores had significantly lower risk of PAD compared to those with inadequate scores, with hazard ratios of 0.37 (95% CI: 0.22–0.61) and 0.25 (95% CI: 0.10–0.65), respectively, in the fully adjusted model. Furthermore, each one-unit increase in the LS7 score was monotonically associated with an additional 22% reduction in PAD risk (HR: 0.78; 95% CI: 0.68–0.90), suggesting a clear linear dose–response relationship between cardiovascular health metrics and PAD incidence. A test for linear trends was calculated, which was significant (*p* < 0.001) for both multivariate adjustment models. No change was observed when we additionally adjusted for type of work (HR per one-unit = 0.79, 95% CI 0.69–0.91).

As shown in Table 4, being never-smoker or a quitter for more than 12 months, being physically active, and having lower baseline glucose levels were associated with a reduced risk of incident PAD. No significant associations were observed for the other LS7 metrics.

Finally, from the updated meta-analysis with four studies (Figure 2), the pooled HR for each additional point in the LS7 score was 0.81 (95% CI: 0.77–0.86), with a moderate, but not statistically significant, heterogeneity (I^2^ = 42.02%, *p* = 0.159). In the leave-one-out meta-analysis, heterogeneity dropped to 0% when we omitted the study by Garg et al.

## 4. Discussion

In this study, adherence to the LS7 was inversely and significantly associated with the incidence of PAD. We found that in participants of the PREDIMED study, a greater compliance with the AHA’s LS7 [8] recommendations was associated with a lower risk of PAD. We observed a 75% (95% CI 35–90%) relative risk reduction of PAD in participants with optimal compliance as compared to those with inadequate LS7 metrics, after adjusting for other risk factors associated with PAD. Moreover, in the updated meta-analysis with four studies, we found a 19% reduction per each point of the LS7 score.

The LS7 metric has been used in numerous studies to show the synergistic effect that the combination of a series of cardiovascular risk factors has in the prevention of cardiovascular diseases [9]. Four studies have specifically analyzed the association between LS7 and risk of PAD [16,17,18,19]. In a cross-sectional analysis with participants from the Jackson Heart Study, of the 4403 participants, those with three or more poor health indicators of LS7 showed a greater probability of prevalent PAD [multivariable OR = 1.30; 95% CI (1.07–1.58)] compared to those with two or fewer poor health indicators [16]. The other three prospective studies were included in the previous meta-analysis [9], two of which were conducted in the US [17,18] and one in China [19]. The strongest association was found in the ARIC (Atherosclerosis Risk in Communities) Study, with a 91% relative risk reduction in participants with optimal LS7 (10–14 points) vs. inadequate LS7 (0–4 points) [17]. A slightly lower reduction was found in the Multi-Ethnic Study of Atherosclerosis (MESA), with 76% (95% CI 41–90%) in those with optimal (12–14 points) vs. inadequate LS7 (0–7 points) [18]. Finally, a non-significant risk reduction was observed in the Asymptomatic Polyvascular Abnormalities Community study (APAC) from China [19].

Our meta-analysis, incorporating our results alongside those three prospective studies, supports a robust and consistent inverse association with a pooled hazard ratio of 0.81 (95% CI: 0.77–0.86), underscoring the generalizability of the LS7 metrics’ protective role against PAD across diverse populations. Notably, our study is the first conducted within a Mediterranean dietary context, adding critical evidence that this association persists in populations with distinct dietary patterns. In addition, the definition of a healthy diet that we used is the Mediterranean diet, which is the nutritional model best supported by randomized trials with hard cardiovascular end-points [12,20]. We observed an inverse association between the MEDAS score and PAD, although it was not statistically significant. This result may be underestimated, as we only considered dietary adherence at baseline, in contrast to the larger effect previously observed in the Mediterranean Diet randomized intervention [13]. Interestingly, we did not observe an association between the LS7 metric of total cholesterol and PAD risk. A significant association was observed in the ARIC study [17] but not in the other studies included in our updated meta-analysis [18,19]. This lack of association may be explained by the fact that our population was older compared to this study [17], and that approximately 70% of our participants already had hypercholesterolemia. These differences between populations may account for the reduction in heterogeneity to 0% when that study [17] was excluded in the leave-one-out meta-analysis. Finally, since physical activity is one of the strongest LS7 metrics associated with a lower PAD risk, further research is needed to explore how different types of physical activity, as well as varying levels of intensity, may modulate this protective effect in combination with the MedDiet or other healthy dietary patterns.

The inverse association between higher LS7 adherence and lower PAD risk can be explained by several interconnected biological mechanisms. Poor cardiovascular health metrics, such as hypertension, dyslipidemia, hyperglycemia, and smoking, contribute directly to endothelial dysfunction, reduced nitric oxide bioavailability, and increased oxidative stress, all key drivers of atherosclerotic plaque formation and peripheral arterial narrowing [21,22]. Chronic low-grade inflammation, reflected by elevated pro-inflammatory cytokines and biomarkers like C-reactive protein, further accelerates arterial injury and promotes plaque instability, increasing the risk of PAD [23]. Importantly, in the Mediterranean context of the PREDIMED study, adherence to a diet rich in unsaturated fats, antioxidants, and polyphenols may offer additional vascular benefits, reducing oxidative damage, improving endothelial function, and lowering thrombogenic potential [24]. In fact, in a small subsample of the PREDIMED trial, we observed a significant reduction of 10 plasma inflammatory biomarkers related to atherosclerosis after 3 years of follow-up [25]. Together, these mechanisms provide a plausible biological explanation for the consistent inverse association between LS7 adherence and PAD observed across diverse populations.

We acknowledge that our study has some limitations. The main limitation is the low number of incident cases of PAD. Despite this limitation, a significant association could be found between LS7 and the incidence of PAD. The lack of repeated measurements of LS7 metrics is another limitation of this study. However, since participants were receiving advice to follow a MedDiet or low-fat diet, improvements in the LS7 are expected, and therefore, even higher risk reductions may be observed compared to our results. In addition, the generalizability of our findings is limited because all our participants lived in a Mediterranean country and were at high cardiovascular risk. The main strengths of this study are the large sample size, the long term of follow-up, and the inclusion of many potential confounders.

Future studies should aim to include larger sample sizes and more diverse populations to strengthen the evidence on the protective role of LS7 adherence against PAD. Our findings suggest that recommending LS7 adherence to high cardiovascular-risk populations may offer added benefits by also reducing the risk of developing PAD, a condition often underrecognized in prevention strategies. Longitudinal studies and intervention trials specifically targeting PAD end-points are needed to confirm these associations and explore whether improving LS7 metrics over time leads to meaningful reductions in PAD incidence. Additionally, integrating LS7-based recommendations into clinical practice guidelines for individuals with high cardiovascular risk may provide a practical, evidence-based approach to simultaneously reduce both coronary and peripheral vascular disease burden.

## 5. Conclusions

In this study, we demonstrate that greater adherence to LS7 metrics is significantly associated with a reduced risk of incident PAD in a Mediterranean population at high cardiovascular risk. These findings extend the known cardiovascular benefits of LS7 to PAD prevention and suggest that encouraging LS7 adherence may serve as an effective strategy for lowering PAD incidence. Integrating LS7-based recommendations into dietary and lifestyle interventions may provide substantial vascular health benefits. Future large-scale, longitudinal studies are warranted to confirm these associations and to assess the long-term impact of LS7 adherence on PAD prevention across diverse populations.

## Figures and Tables

**Figure 1 nutrients-17-02058-f001:**
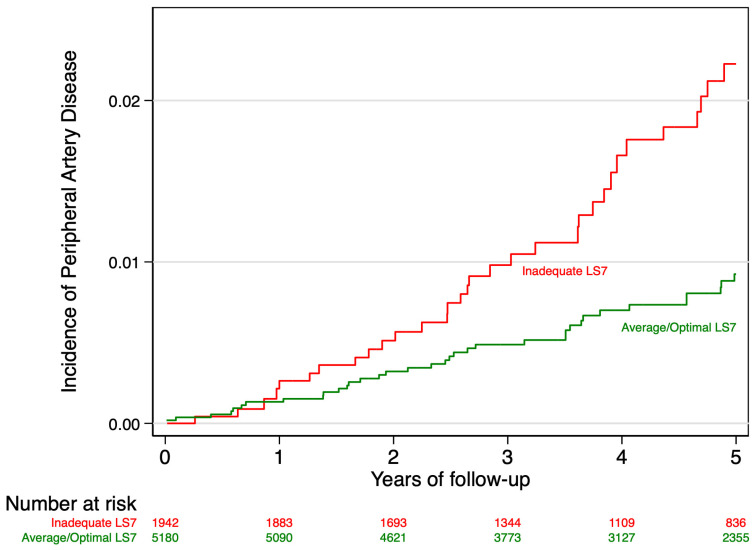
Nelson–Aalen estimator of the cumulative risk of suffering PAD according to the metrics as a dichotomous variable. Adjusted for age and sex with inverse probability weighting.

**Figure 2 nutrients-17-02058-f002:**
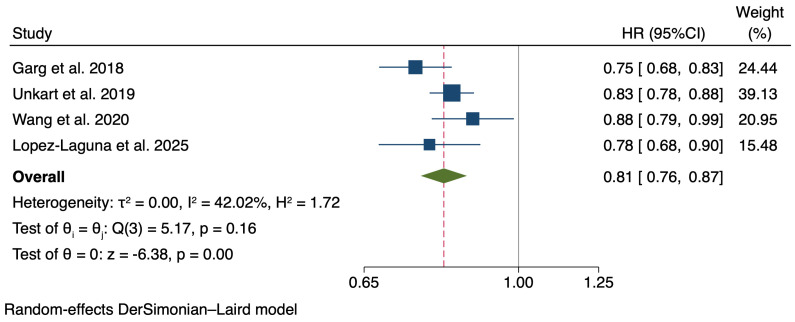
Forest plot showing the association between Life’s Simple 7 (for each additional point) and peripheral artery disease risk. Estimates are from previous studies [17,18,19] and from the present analysis.

**Table 1 nutrients-17-02058-t001:** Life’s Simple 7 components and scoring criteria.

Component	Inadequate (0 Points)	Average (1 Point)	Optimal (2 Points)
**1. Smoking Status**	Current smoker	Former smoker who quit within the past year	Never smoker or former smoker who quit > 1 year ago
**2. Body Mass Index (BMI)**	≥30 kg/m^2^	25–29.9 kg/m^2^	<25 kg/m^2^
**3. Physical Activity**	≤10 METs-min/wk	11–239 METs-min/wk	≥240 METs-min/wk
**4. Blood Pressure**	Systolic BP ≥ 140 mmHg or diastolic BP ≥ 90 mmHg	Systolic BP 120–139 mmHg or diastolic BP 80–89 mmHg	Systolic BP < 120 mmHg and diastolic BP < 80 mmHg, without antihypertensive medication
**5. Total Cholesterol**	≥240 mg/dL	200–239 mg/dL	<200 mg/dL, without lipid-lowering medication
**6. Glucose Metabolism**	Fasting glucose ≥ 120 mg/dL	Fasting glucose 100–119 mg/dL	Fasting glucose < 100 mg/dL, without insulin or oral hypoglycemic medication
**7. Mediterranean Diet Adherence**	Low adherence (MEDAS score ≤ 6)	Moderate adherence (MEDAS score 7–9)	High adherence (MEDAS score ≥ 10)

Note: The total LS7 score ranges from a minimum of 0 points to a maximum of 14 points, with higher scores indicating better cardiovascular health.

**Table 2 nutrients-17-02058-t002:** Characteristics of the study participants by Life’s Simple 7 categories.

	Inadequate (0–5)	Average (6–8)	Optimal (9–14)
N	1941	4173	1008
Age. years (SD)	66.4 (6.5)	67.1 (6.1)	67.3 (6.1)
Sex (% Women)	1007 (51.9)	2507 (60.1)	581 (57.6)
Level of studies (% Primary or less)	1511 (77.9)	3286 (78.7)	743 (73.7)
Work status			
Employed	288 (14.8)	491 (11.8)	115 (11.4)
Unpaid domestic work	571 (29.4)	1402 (33.6)	315 (31.3)
Retired	1004 (51.7)	2150 (51.5)	555 (55.1)
Other	78 (4.0%)	130 (3.1%)	23 (2.3%)
LS7 metrics (SD)	4.3 (0.9)	6.7 (0.8)	9.4 (0.6)
Total cholesterol (mg/dL) (SD)	208.0 (38.6)	212.4 (34.1)	209.8 (29.8)
SBP (mmHg) (SD)	153.9 (18.2)	148.3 (18.6)	140.0 (17.9)
DBP (mmHg) (SD)	84.4 (9.9)	82.8 (10.0)	79.6 (9.9)
Glucose (mg/dL) (SD)	139.0 (41.2)	118.6 (37.1)	103.4 (28.5)
Hypercholesterolemia (%)	1517 (78.2)	2920 (70.0)	701 (69.5)
Hypertension (%)	1601 (82.5)	3489 (83.6)	799 (79.3)
Statins (%)	877 (44.6)	1614 (38.7)	380 (37.7)
Cardiovascular treatments * (%)	1406 (72.4)	2926 (70.1)	650 (64.5)
Physical activity (METs-min/d) (SD)	146.3 (192.0)	242.2 (237.8)	350.6 (268.9)
Body Mass Index (SD)	31.7 (3.7)	29.8 (3.7)	27.3 (2.9)
MEDAS (0 to 14 points) (SD)	7.7 (1.9)	8.8 (1.8)	9.9 (1.6)
Smoking (%)			
Never	934 (48.1)	2753 (66.0)	682 (67.7)
Current smoker	587 (30.2)	384 (9.2)	23 (2.3)
Former smoker	420 (21.6)	1036 (24.8)	303 (30.1)
Height. meters (SD)	1.61 (0.1)	1.60 (0.1)	1.60 (0.1)
Waist circumference (centimeters) (SD)	105.3 (9.7)	99.7 (10.2)	94.1 (9.5)
Total energy intake (Kcal/d) (SD)	2235 (589)	2240 (545)	2297 (518)
Alcohol (g/d) (SD)	9.4 (16.0)	8.1 (13.8)	7.7 (11.7)

SD: standard deviation; LS7: Life’s Simple 7; SBP: systolic blood pressure; DBP: diastolic blood pressure; METs: metabolic equivalents; BMI: body mass index; MEDAS: Mediterranean Diet adherence screener. * Cardiovascular treatment: antiplatelet agents (aspirin), diuretics, angiotensin-converting enzyme inhibitors or angiotensin receptor blockers, and beta-blockers.

**Table 3 nutrients-17-02058-t003:** Associations of Life’s Simple 7 categories with incident PAD.

	Life’s Simple 7 Categories			
	Inadequate (0–5)	Average (6–8)	Optimal (9–14)	Per 1-Unit Increase
**N**	1941	4173	1008	
**Cases of PAD**	40	40	7	
**Incidence rate (95% CI), 1000 p-yr**	4.81 (3.53–6.55)	2.16 (1.58–2.94)	1.57 (0.75–3.30)	
**Crude HR (95% CI)**	1 (ref)	0.45 (0.29–0.69)	0.33 (0.15–0.73)	0.82 (0.73–0.92)
**Adjusted HR (95% CI), model 1**	1 (ref)	0.40 (0.25–0.64)	0.29 (0.12–0.67)	0.80 (0.72–0.90)
**Adjusted HR (95% CI), model 2**	1 (ref)	0.37 (0.22–0.61)	0.25 (0.10–0.65)	0.78 (0.68–0.90)

Model 1 adjusted for age, sex, education, and alcohol intake, and stratified by intervention group and recruitment center. Model 2 additionally adjusted for height, waist, family history of CVD, diabetes status, statin use, and energy.

**Table 4 nutrients-17-02058-t004:** Associations of individual Life’s Simple 7 components with incident PAD.

Life’s Simple 7 Component	Cases/N	HR (95% CI)
**Smoking**		
Inadequate (current smoker)	26/994	1 (ref)
Average (quit ≤ 12 months)	4/197	0.78 (0.27–2.24)
Optimum (never/quit > 12 months)	57/5931	0.53 (0.32–0.86)
**Body Mass Index**		
Inadequate (≥30)	34/3344	1 (ref)
Average (25−29.9)	43/3240	0.97 (0.61–1.55)
Optimum (<25)	10/538	1.47 (0.70–3.09)
**MEDAS score**		
Inadequate (0−6 points)	15/1096	1 (ref)
Average (7–10 points)	53/3993	0.89 (0.49–1.59)
Optimum (10–14 points)	19/2033	0.70 (0.34–1.42)
**Physical activity**		
Inadequate (<10 METs-h/d)	9/966	1 (ref)
Average (10 to <240 METs-h/d)	52/3592	0.94 (0.45–1.96)
Optimum (≥240 METs-h/d)	26/2564	0.44 (0.20–0.98)
**Total cholesterol**		
Inadequate (≥240 mg/dL)	18/1322	1 (ref)
Average (200−239 mg/dL)	37/3148	1.01 (0.57–1.81)
Optimum (<200 mg/dL)	18/929	1.15 (0.57–2.34)
**Glucose**		
Inadequate (≥120 mg/dL)	52/2941	1 (ref)
Average (100 to <120 mg/d)	16/1571	0.48 (0.27–0.85)
Optimum (<100 mg/dL)	16/2448	0.35 (0.20–0.63)
**Blood pressure**		
Inadequate (SBP ≥ 140 or DBP ≥ 90 mmHg)	63/4816	1 (ref)
Average (SBP 120−139 mmHg, DBP 80−89 mmHg)	21/1981	1.06 (0.65–1.75)
Optimum (SBP < 120 mmHg and DBP < 80 mmHg)	1/159	0.64 (0.08–4.86)

Models adjusted for age, sex, education, and alcohol intake, and stratified by intervention group and recruitment center. MEDAS: Mediterranean Diet adherence screener.

## Data Availability

The data generated and analysed in this research are not publicly accessible due to national regulatory requirements and ethical considerations, as certain information could potentially compromise the consent provided by study participants, who authorized data use exclusively by the original research team. Nevertheless, access to these data may be granted through execution of a data sharing agreement that requires approval from the appropriate institutional review boards and the PREDIMED study steering committee.

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
