# Peer review of "Life’s Simple 7 and Risk of Peripheral Artery Disease: Results from the PREDIMED Study and an Updated Meta-Analysis"

_nutrients, 2025, doi:10.3390/nu17132058_

Round 1
Reviewer 1 Report
Comments and Suggestions for Authors
The idea behind the study is nice, to gain new evidences regarding the importance of RF control in primary prevention of ACVD, namely PAD. The methodology, the results are convincing, although the results being somewhat those expected. Just a few questions, I would like to ask the authors:
- how was the PAD diagnosed, just by symptoms? - any ABI, angiography, Dupplex Doppler study?
- the inflammatory status of the patients?
- any role of mediterranian diet in prevention of PAD?
- why cholesterol level was not related to incident PAD
- statin usage was evaluated?
Author Response
Reviewer 1.
The idea behind the study is nice, to gain new evidences regarding the importance of RF control in primary prevention of ACVD, namely PAD. The methodology, the results are convincing, although the results being somewhat those expected.
RESPONSE: Thank you for your comment. We agree that the main message in our manuscript is to confirm the results from previous studies in a Mediterranean population and to update the previous meta-analysis. Please, find below our answers to your specific questions. This has been very helpful to improve our manuscript.
Just a few questions, I would like to ask the authors:
- how was the PAD diagnosed, just by symptoms? - any ABI, angiography, Dupplex Doppler study?
RESPONSE: We have expanded the information about the adjudication of PAD confirmed cases:
“Medical records were examined, blinded to the exposure, by the Clinical Event Adjudication Committee to document the presence of PAD diagnosis if participants reported a disorder of the lower extremity arteries during follow-up. The information to adjudicate new cases of PAD included ankle-brachial index measurements, treadmill exercise test, limb segmental pressure measurements, pulse volume recordings and imaging tests (duplex ultrasonography, magnetic resonance angiography, computed tomographic angiography or catheter-based radiocontrast angiography). Confirmed PAD cases required at least one of the following criteria: an ABI lower than 0.9 at rest, an objective evidence of arterial occlusive disease, or an endovascular or open surgical revascularization (or amputation)” (Lines 133-143, Pages 3-4).
- the inflammatory status of the patients?
RESPONSE: Unfortunately, we only have this information from a small subsample of PREDIMED participants (n=930). In this subsample the following inflammation markers were measured: IL-6, interleukin-6; ICAM-1, intercellular adhesion molecule-1; hsCRP, highly sensitive C-reactive protein; VCAM-1, vascular cell adhesion molecule-1 (1). In a more recent analysis 285 participants in the PREDIMED trial, 14 plasma inflammatory biomarkers related to atherosclerosis were measured at baseline and after 3 years of follow-up (2). A significant reduction of 10 of these biomarkers was observed in the Mediterranean diet groups suggesting that the anti-atherogenesis is partly mediated by these anti-inflammatory mechanisms.
References:
1) Salas-Salvadó J, Garcia-Arellano A, Estruch R, Marquez-Sandoval F, Corella D, Fiol M, et al. Components of the Mediterranean-type food pattern and serum inflammatory markers among patients at high risk for cardiovascular disease. Eur J Clin Nutr. 2008 May;62(5):651-9. doi: 10.1038/sj.ejcn.1602762.
2) Urpi-Sarda M, Casas R, Sacanella E, Corella D, Andrés-Lacueva C, Llorach R, Garrabou G, Cardellach F, Sala-Vila A, Ros E, Ruiz-Canela M, Fitó M, Salas-Salvadó J, Estruch R. The 3-Year Effect of the Mediterranean Diet Intervention on Inflammatory Biomarkers Related to Cardiovascular Disease. Biomedicines. 2021 Jul 22;9(8):862. doi: 10.3390/biomedicines9080862.
We have added the following sentence in the new version of the manuscript.
“In fact, in a small subsample of the PREDIMED trial, we observed a significant reduction of 10 plasma inflammatory biomarkers related to atherosclerosis after 3 years of follow-up [25].” (Lines 347-350, page 12)
- any role of Mediterranean diet in prevention of PAD?
RESPONSE: Thank you for this question. We published in 2014 (reference 13 in the manuscript) the effect of the Mediterranean diet (MedDiet) on PAD incidence in participants of the PREDIMED trial. We observed a relative risk reduction of 66% (95% CI 42%-80%) for PAD in the MedDiet group enriched with extra-virgin olive oil and a 50% (95% CI 19%-70%) relative reduction in the MedDiet group enriched with nuts as compared to the control group. This is the largest effect of the MedDiet intervention observed in the PREDIMED trial, compared to other main outcomes (cardiovascular disease, diabetes, or atrial fibrillation). However, the confidence intervals were wide. The effect observed in our present analysis was a non-significant 30% risk reduction for PAD in participants with an optimum MEDAS score (10–14 points) compared to an inadequate score (0–6 points). This smaller effect, compared to the results of the randomized intervention mentioned before, can be explained by the observational nature of our analysis and the potential existence of residual confounding. In addition, the high-quality of the extravirgin olive oil, very rich in phenolics, that was provided in the arm of the trial randomized to MedDiet plus free provision of extravirgin olive oil is not captured by the MEDAS score. In any case, the 30% relative reduction now observed for a high MEDAS is included in the confidence nterval for the randomized estimate that did not include the free provision of extravirgin olive oi.
We have added this sentence in the Discussion:
“Notably, our study is the first conducted within a Mediterranean dietary context, adding critical evidence that this association persists in populations with distinct dietary patterns. In addition, the definition of a healthy diet that we used is the Mediterranean diet, which is the nutritional model best supported by randomized trials with hard clinical cardiovascular end-points [12, 20]. We observed an inverse association between the MEDAS score and PAD, although it was not statistically significant. This result may be underestimated, as we only considered dietary adherence at baseline, in contrast to the larger effect previously observed in the MedDiet randomized intervention [13]” (Lines 315-324, page 11)
- why cholesterol level was not related to incident PAD
RESPONSE: Thank you for this raising this issue. Two previous studies included in our meta-analysis (Unkart et al, and Wang et al) did not find any association between total cholesterol categories and PAD risk. However, Garg et al observed a significant association in the comparison between ideal vs poor total cholesterol levels (multivariable HR=0.78 (95%CI 0.61-0.98) among ARIC participants. The population in the ARIC cohort was younger and with lower prevalence of cardiovascular risk factors compared to the PREDIMED population. In the PREDIMED trial, around 70% of the participants had hypercholesterolemia and probably this explains the lack of association between the total cholesterol metric of the LS7 and PAD. Given that the requirement for including non-diabetic participants in the trial was to have at least 3 major cardiovascular factors, the presence of hypercholesterolemia tended to be inversely related to smoking or hypertension, that are stronger risk factors. This
In the new version of the manuscript we have included the information about the percentage of hypercholesterolemia and hypertension in Table 1:
|
Hypercholesterolemia (%) |
1517 (78.2) |
2920 (70.0) |
701 (69.5) |
|
Hypertension (%) |
1601 (82.5) |
3489 (83.6) |
799 (79.3) |
We have also added a sentence in the Discussion:
“Interestingly, we did not observe any association between the LS7 metric of total cholesterol and PAD risk. A significant association was observed in the ARIC study [17] but not in the other studies included in our updated meta-analysis [18, 19]. This lack of association may be explained by the fact that our population was older compared to this study [17], and that approximately 70% of PREDIMED participants already had hypercholesterolemia” (Lines 324-329, page 11).
- statin usage was evaluated?
RESPONSE: Thank for this question. We have added this information in Table 1, and we have also added the use of other cardiovascular pharmacological treatments.
|
Statins (%) |
877 (44.6) |
1614 (38.7) |
380 (37.7) |
|
Any cardiovascular drug use* (%) |
1406 (72.4) |
2926 (70.1) |
650 (64.5) |
In addition, we have adjusted for statin use in the multivariable analysis. The results have slightly improved showing a stronger association between LS7 and risk of PAD:
Results from the previous analysis (without adjusting for statin use, Table 3):
|
Adjusted HR (95% CI), model 2 |
1 (ref) |
0.41 (0.25-0.68) |
0.29 (0.11-0.75) |
0.82 (0.72-0.93) |
Results from the previous analysis (adjusting for statin use, Table 3, new version of the manuscript):
|
Adjusted HR (95% CI), model 2 |
1 (ref) |
0.37 (0.22-0.61) |
0.25 (0.10-0.65) |
0.78 (0.68-0.90) |
We have also modified the updated meta-analysis with this new HR although there was only a slight change in the confidence intervals of the pooled HR, but the point estimate did not change (HR: 0.81, 95% CI 0.77-0.86 in the old version and 0.81, 95% CI 0.76-0.87 in the revised version)

Reviewer 2 Report
Comments and Suggestions for Authors
This study presents a valuable contribution to the understanding of peripheral artery disease (PAD) and its association with Life’s Simple 7 (LS7) metrics, particularly within the Mediterranean population. Its thorough documentation of participants' cardiovascular health and its integration with previous epidemiological findings highlight the relevance of LS7 adherence in PAD prevention. Moreover, the study’s methodological approach, including multivariate Cox regression models and meta-analysis, provides a comprehensive overview of the statistical robustness behind these conclusions. However, some points merit further consideration:
- The study primarily categorizes cardiovascular health based on LS7 metrics. While this approach is well-established, could a more granular analysis of individual components (e.g., physical activity intensity, dietary patterns beyond MEDAS) provide additional insights into specific therapeutic effects and potential toxicities of these interventions?
- The reliance on a high-risk Mediterranean population may limit the generalizability of the conclusions to broader populations. Would including additional cohorts from non-Mediterranean regions or those with varying baseline risk factors enhance the applicability and robustness of the conclusions drawn?
- The paper suggests that greater LS7 adherence is associated with reduced PAD risk. Could further exploration of potential confounding factors (such as socioeconomic status, medication adherence, or genetic predisposition) through sensitivity analyses strengthen the causal interpretation of this relationship?
- The meta-analysis supports the inverse association between LS7 adherence and PAD risk, yet variability across included studies remains moderate. Would a stratified analysis examining differences in dietary components, lifestyle interventions, or healthcare access provide further clarity on the sources of heterogeneity?
5. Given that LS7 is assessed at baseline, the study does not account for potential improvements or declines in cardiovascular health over time. Could an analysis incorporating repeated measures of LS7 metrics elucidate whether sustained adherence or improvements confer greater long-term benefits in PAD prevention?
Author Response
Reviewer 2
This study presents a valuable contribution to the understanding of peripheral artery disease (PAD) and its association with Life’s Simple 7 (LS7) metrics, particularly within the Mediterranean population. Its thorough documentation of participants' cardiovascular health and its integration with previous epidemiological findings highlight the relevance of LS7 adherence in PAD prevention. Moreover, the study’s methodological approach, including multivariate Cox regression models and meta-analysis, provides a comprehensive overview of the statistical robustness behind these conclusions.
RESPONSE: Thank you for your comments which have helped to improve our manuscript. We have answered to each point below.
However, some points merit further consideration:
- The study primarily categorizes cardiovascular health based on LS7 metrics. While this approach is well-established, could a more granular analysis of individual components (e.g., physical activity intensity, dietary patterns beyond MEDAS) provide additional insights into specific therapeutic effects and potential toxicities of these interventions?
RESPONSE: We agree that it will be interesting to further explore the role that not only the amount of physical activity but also the intensity and type of physical activity, as well as other dietary patterns may have on the reduction of PAD incidence. We already have addressed the MEDAS dietary pattern and went deeper on this association when responding to another reviewer. We feel that further inclusion of other more granular analyses will considerably expand the manuscript beyond the allowable limits and, perhaps, they should deserve a separate manuscript.
We have added the following sentence in the Discussion:
“Finally, since physical activity is one of the strongest LS7 metrics associated with a lower PAD risk, further research is needed to explore how different types of physical activity, as well as varying levels of intensity, may modulate this protective effect in combination with the MedDiet or other healthy dietary patterns” (Lines 330-34, pages 11-12)
- The reliance on a high-risk Mediterranean population may limit the generalizability of the conclusions to broader populations. Would including additional cohorts from non-Mediterranean regions or those with varying baseline risk factors enhance the applicability and robustness of the conclusions drawn?
RESPONSE: Thank you for raising a very relevant question in terms of the applicability and transferability of our results. Although we used a Mediterranean population at high risk of cardiovascular disease, our updated meta-analysis including results from other countries may increase the robustness of the results and its generalizability. However, the number of studies is still small and further studies from other regions and with different baseline cardiovascular risk factors are needed. We already mentioned this in our discussion (lines 330-337)
- The paper suggests that greater LS7 adherence is associated with reduced PAD risk. Could further exploration of potential confounding factors (such as socioeconomic status, medication adherence, or genetic predisposition) through sensitivity analyses strengthen the causal interpretation of this relationship?
RESPONSE: Thank you for this interesting question. We have additionally adjusted for statin use, following the comment from another reviewer. Following your suggestion, we have also adjusted for type of work and no change was observe compared to the fully adjusted multivariable model.
We have added the following sentences:
Methods:
“As an ancillary analysis we also adjusted for type of work (employed, unpaid domestic work, retired, and other).” (Lines 190-192, page 5).
Results:
“No change was observed when we additionally adjusted for type of work (HR per one-unit=0.79, 95%CI 0.69-0.91).” (Lines 265-6, page 8)
- The meta-analysis supports the inverse association between LS7 adherence and PAD risk, yet variability across included studies remains moderate. Would a stratified analysis examining differences in dietary components, lifestyle interventions, or healthcare access provide further clarity on the sources of heterogeneity?
RESPONSE: Thank you, this is is a relevant question but the small number of studies included in this analysis limits the exploration of potential sources of heterogeneity. However, we have repeated using the leave-one-out option to explore which study explains the larger percentage of heterogeneity.
We found that the largest change in effect size happened when the omitted study was the paper published by Garg et al, as it is shown in the following figure:
In the meta-analysis the I2 was 42%. When we excluded the study by Garg et al, the I2 was 0. The I2 changes were smaller or increased when we omitted the other studies, one by one (55% when Unkart et al was omitted, 36% when Wang was omitted, and 57.8% when Lopez-Laguna was omitted).
We have added the following sentences:
Methods:
“We also conducted a leave-one-out sensitivity analysis to evaluate the contribution of each study to the overall heterogeneity” (Lines 202-3, page 5).
Results:
“In the leave-one-out meta-analysis, heterogeneity dropped to 0% when we omitted the study by Garg et al” (Lines 282-3, page 10)
Discussion:
“This lack of association may be explained by the fact that our population was older compared to this study [17], and that approximately 70% of our participants already had hypercholesterolemia. These differences between populations may account for the reduction in heterogeneity to 0% when that study [17] was excluded in the leave-one-out meta-analysis”. (Lines 327-331, page 11)
- Given that LS7 is assessed at baseline, the study does not account for potential improvements or declines in cardiovascular health over time. Could an analysis incorporating repeated measures of LS7 metrics elucidate whether sustained adherence or improvements confer greater long-term benefits in PAD prevention?
RESPONSE: Thank you for your comment. We agree that the type of analysis that you suggested is an interesting option, however the main limitation is the existence of substantial amounts of missing data for some of the variables used to build LS7 during follow-up. In the PREDIMED study, data were collected annually, and it would indeed be valuable to analyze repeated measurements of LS7. However, we believe this additional analysis falls outside the scope of the current manuscript submitted to this Special Issue and would considerably increase its length and will be complicated and unstable given the amount of missing values. This could be the focus of a future manuscript, in which we could examine in detail how improvements in overall LS7 scores—and in each individual metric—during follow-up are associated with long-term benefits in PAD prevention. Different imputation methods can be applied to account for missing data during follow-up. Additionally, it would be of interest to explore potential interactions with sex, age, and the intervention groups of the PREDIMED trial. In this context, an overall improvement in LS7 metrics is expected, which may translate into an even greater reduction in PAD risk.
To address this point, we have added a new limitation at the end of the Discussion section:
“The lack of repeated measurement of LS7 metrics is another limitation of this study. However, since participants were receiving an advice to follow a MedDiet or low-fat diet, improvements in the LS7 are expected and therefore even higher risk reductions may be observed compared to our results.” (Lines 355-359, page 12)
